# 4-aminopyridine improves evoked potentials and ambulation in the *taiep* rat: A model of hypomyelination with atrophy of basal ganglia and cerebellum

Jose R. Eguibar[1,2]*, Carmen Cortes[1]*, Victor H. Hernandez[3], Alejandra Lopez-Juarez[3], Valeria Piazza[4], Diego Carmona[3,4], Anke Kleinert-Altamirano[5], Blanca Morales-Campos[6], Emilio Salceda[7], Manuel Roncagliolo[8]

1 Laboratorio de Neurofisiología de la Conducta y Control Motor, Instituto de Fisiología, Benemérita Universidad Autónoma de Puebla, Puebla, Pue, México, 2 Dirección General de Desarrollo Internacional, Benemérita Universidad Autónoma de Puebla, Puebla, Pue, México, 3 Departamento de Ingenierías Química, Electrónica y Biomédica, División de Ciencias e Ingenierías, Universidad de Guanajuato, León, Gto, México, 4 Centro de Investigaciones en Óptica, A.C., León, Gto, México, 5 Centro de Rehabilitación Infantil Teletón, Chiapas, Chis, México, 6 Departamento de Fisiología, Facultad de Medicina, Benemérita Universidad Autónoma de Puebla, Puebla, Puebla, Pue, México, 7 Revista Elementos, Benemérita Universidad Autónoma de Puebla, Puebla, Pue, México, 8 Instituto de Fisiología, Facultad de Ciencias, Universidad de Valparaíso, Valparaíso, Chile

* jose.eguibar@correo.buap.mx (JRE); carmen.cortes@correo.buap.mx (CC)

**Data Availability Statement:** The data underlying the results presented in the study are available

## Abstract

The *taiep* rat is a tubulin mutant with an early hypomyelination followed by progressive demyelination of the central nervous system due to a point mutation in the *Tubb4a* gene. It shows clinical, radiological, and pathological signs like those of the human leukodystrophy hypomyelination with atrophy of the basal ganglia and cerebellum (H-ABC). *Taiep* rats had tremor, ataxia, immobility episodes, epilepsy, and paralysis; the acronym of these signs given the name to this autosomal recessive trait. The aim of this study was to analyze the characteristics of somatosensory evoked potentials (SSEPs) and motor evoked potentials (MEPs) in adult *taiep* rats and in a patient suffering from H-ABC. Additionally, we evaluated the effects of 4-aminopyridine (4-AP) on sensory responses and locomotion and finally, we compared myelin loss in the spinal cord of adult *taiep* and wild type (WT) rats using immunostaining. Our results showed delayed SSEPs in the upper and the absence of them in the lower extremities in a human patient. In *taiep* rats SSEPs had a delayed second negative evoked responses and were more susceptible to delayed responses with iterative stimulation with respect to WT. MEPs were produced by bipolar stimulation of the primary motor cortex generating a direct wave in WT rats followed by several indirect waves, but *taiep* rats had fused MEPs. Importantly, *taiep* SSEPs improved after systemic administration of 4-AP, a potassium channel blocker, and this drug induced an increase in the horizontal displacement measured in a novelty-induced locomotor test. In *taiep* subjects have a significant decrease in the immunostaining of myelin in the anterior and ventral funiculi of the lumbar spinal cord with respect to WT rats. In conclusion, evoked potentials are useful to evaluate myelin alterations in a leukodystrophy, which improved after systemic administration of 4-

from Institue of Physiology, Benémerita Universidad Autónoma de Puebla. http://148.228.21.2/sistemas/Eguibar_laboratory/.

**Funding:** First: Consejo Nacional de Ciencia y Tecnología (CONACYT) through Programas Nacionales Estratégicos (PRONACES) number 194171 being the recipient Dr. Victor Hugo Hernandez (VHH). Second: Vicerrectoría de Investigación y Estudios de Posgrado (VIEP) de la Benemérita Universidad Autónoma de Puebla (BUAP) grant to Cuerpo Académico en Neuroendocrinología (CA-BUAP-288) 2023 to Jose R. Eguibar and Carmen Cortes.

**Competing interests:** The authors have declared that no competing interest exist.

AP. Our results have a translational value because our findings have implications in future medical trials for H-ABC patients or with other leukodystrophies.

## Introduction

Somatosensory and motor-evoked potentials allow rapid clinical evaluation of the neurophysiological conditions in afferent and efferent pathways. They are an excellent tool to evaluate the physiological state of axons and their myelin sheaths. In the case of myelin disorders, changes in the evoked potentials allow to analyze the state of the somatosensory and motor tracts involved in the central transmission of different sensory and motor modalities [1, 2].

The *taiep* rat shows a progressive motor syndrome characterized by tremor, ataxia, immobility episodes, epilepsy, and paralysis. It was obtained at the Institute of Physiology of the Benemérita Universidad Autónoma de Puebla, México, as a spontaneous recessive tubulin mutation during the inbreeding process to obtain a subline of Sprague-Dawley with high-yawning frequency [3, 4]. The genetic change causes an initial hypomyelination followed by a progressive demyelination exclusively in the central nervous system (CNS) with a somatic recessive inheritance [3, 5, 6]. Recently, our research group demonstrated that *taiep* rats have a point mutation in *Tubb4a* gen and its corresponding protein TUBB4A (OMIM 602662), with pathological characteristics similar to the human leukodystrophy hypomyelination with atrophy of the basal ganglia and cerebellum (H-ABC); [7, 8]. TUBB4A is highly expressed in the CNS, particularly in the cerebellum and white matter tracts, with moderate expression in the striatum [9]. It is relevant that TUBB4A has the highest expression in mature myelinating oligodendrocytes [10].

Since the first description of seven patients made by Van der Knaap et al., [11], neuroimaging studies using magnetic resonance imaging (MRI) are pivotal to diagnosing H-ABC. However, the clinical diagnosis of leukodystrophies is difficult, and recognizing H-ABC is even more since it is a rare syndrome, still unfamiliar to most physicians. The MRI features of H-ABC, together with the clinical signs, give a strong indication of the disease, and recent advances in whole-exome or whole-genome sequencing techniques provide a definitive diagnosis in 80–90% of patients [9, 12, 13]. This represents an obstacle in low-income countries and therefore the real population of patients affected by H-ABC is likely to be higher than the approximately 200 patients that have been reported so far [14, 15].

Most patients have pyramidal and extrapyramidal symptoms such as tremor, ataxia, and dystonia that tend to worsen with age. It is well recognized that TUBB4A mutations produce a clinical spectrum ranging from only hypomyelination, to H-ABC or dystonia type-4 [15, 16]. Until now, 40 different mutations of TUBB4A have been found, being the p.Asp249Asn the most frequent [15]; it is also the mutation that our group identified in the first Mexican patient diagnosed with H-ABC [8].

We have clearly shown that *taiep* rats share with human H-ABC patients the same magnetic resonance images, genetic and histological features, i.e. in the MRI studies we showed that *taiep* rats suffer hypomyelination in different central tracts such as the corpus callosum and the anterior commissure [7], associated with ventriculomegaly and atrophy of the cerebellum and the caudate nucleus [8, 14]. Furthermore, the *taiep* rat carries the Ala302Thr mutation in TUBB4A [14, 15]. At the ultrastructural level, the main finding is an accumulation of microtubules in the soma and processes of the oligodendrocytes [6], which produces a significant reduction of the g ratio due to thinner myelin sheaths [17]. The damaged myelin produces an

altered compound action potential (CAP) with a significant decrease in the conduction velocity in the *in vitro* optic nerve preparation [18]. The *taiep* CAP has higher sensitivity to potassium channel blocker such as 4-aminopyridine (4-AP) due to paranodal junction alterations [18]. Indeed, by using electron microscopy, we demonstrated that, from 30 postnatal days (PND), the internodal myelin sheath of WT rats are organized into 5 to 8 dense lines, and paranodal loops are attached to the axolemma, both of which persist for the whole life of the animal. On the other hand, in the *taiep* rat, between 15 to 30 PND, axons appear hypomyelinated with no more than 2 to 4 loose dense lines. The myelin degeneration progresses, the paranodal loops detach from the axolemma at 30 PND and axons appear severely demyelinated at 180 PND, with an accumulation of microtubules within oligodendrocyte processes and the paranodal region showing several loops retracted and detached from the axonal membrane [18].

Recently, our research group showed that the central components of auditory evoked potentials in a H-ABC patient have a delayed central response that is similar to that obtained in adult male *taiep* rats [8], and also with a progressive deterioration in the auditory evoked potentials in *taiep* rats [19].

The aims of this study were to analyze first somatosensory and motor evoked potentials in adult *taiep* rats. Second determine the effects on systemic administration of 4-aminopyridine on SSEPs and in the locomotion and tremor; and third measuring myelin immunostaining in the dorsal and ventral columns in the lumbar spinal cord. Four we analyzed somatosensory evoked potentials by upper and lower extremities stimulation in a patient with H-ABC.

## Material and methods

### General procedures

The experiments were performed in 38 male *taiep* and 32 wild type Sprague-Dawley (WT) rats of 3 months of age, supplied by our animal room facility. The subjects were housed in Plexiglas cages (46 x 32 x 20 cm) in groups of 2–3 rats per cage in a room with controlled conditions of temperature (22 ± 2˚C), relative humidity between 30%–45% and a light–dark cycle 12:12 (lights on at 07:00). Balanced rodent pellets (Purina Mills, USA) and purified water Ciel™ (Coca Cola Co., México) were provided *ad libitum*.

All the procedures described were performed in compliance with the Laws and Codes approved in the Seventh title of the Regulations of the General Law of Health regarding Health Research of the Mexican Government (NOM-062-ZOO-1999) and following the NIH Guide for the Care and Use of Laboratory Animals (Eighth edition, 2011). Even though this work does not include any participation or interaction with the patient, permission to use the evoked potentials from the clinical file was granted from the patient family and approved by the institutional committee of bioethics in research of the University of Guanajuato (CIBIUG-P19-2019). All the animal procedures were approved by the institutional committee of bioethics in research of the Benemérita Universidad Autónoma de Puebla Animal Care and Use Committee CICUAL-Proyecto-00255.

For the electrophysiological experiments, the rats were anesthetized with urethane 25% with a dose of 100 mg/Kg, administered by i.p. route. Anesthesia was supplemented during the dissection and along experimental protocol with additional doses to maintain a deep level of anesthesia to avoid noxious responses to tail pinch [20]. The right jugular vein was cannulated with a polyethylene catheter 22 gauge to allow the injection of fluids. The lumbosacral region was exposed by laminectomy; the posterior tibial nerve was stimulated with a pair of Ag/AgCl silver electrodes (World Precision Instruments, USA). The cathode was fixed to the medial part of the Achilles tendon and the anode was attached to the plantar surface.

After the main dissection, the rats were transferred to a Kopf-type stereotaxic instrument which allowed fixation of the head and the spinal vertebrae (Stoelting Co., USA). Fine stainless-steel stimulating electrodes were mounted near the dura in the region of the hindlimb area, in the cerebral cortex following the Paxinos and Watson stereotaxic coordinates [21]. In another group of rats, a large craniotomy was performed, and the dura opened; the reservoirs made with the skin flaps were filled with mineral oil, then the cerebral cortex was exposed and stimulated with Ag/AgCl small ball electrode in the Brodman area 4, and the reference electrode was a platinum needle inserted in the hard palate (Grass-Astromed, USA).

## Recording of the somatosensory evoked potentials

The somatosensory evoked potentials (SSEPs) were induced through a common peroneus nerve stimulation recorded at lower thoracic and lumbar spinal cord segments by means of platinum/iridium needle electrodes (World Precision Instruments, USA) inserted in different interspinous ligaments. The stimulus pulses were delivered through Master-8 equipment (A. M.P.I., Israel) and consisted of 100 μsec, 1–3 mA pulses through electrical isolation units (Iso-flex, A.M.P.I., Israel). Motor-evoked potentials were recorded at different interspinous ligaments. The signals were amplified (2,000 or 5,000 X), band pass filtered from 3 to 3,000 Hz through a Grass amplifier (P511), and the analog signals were digital-converted (Axon-Instruments, USA) and stored in a hard disk of a computer for subsequent offline analysis.

## Motor evoked potentials

Stimulation of the motor cortex was performed according to previously published reports in rats [22–24]. We used constant current stimulation delivered between an insulated Ag/AgCl electrode (anode), around 250 μm in diameter (World Precision Instruments, USA), on the surface of the right motor cortex and another electrode (cathode) placed in the soft palate. Single or iterative stimulus pulses were delivered through Master-8 equipment (A.M.P.I., Israel) and consisted of 100 μsec, 1–3 mA pulses through electrical isolation units (Iso-flex, A.M.P.I., Israel).

The motor-evoked potentials (MEPs) were recorded from the spinal cord with a 500 μm insulated tungsten electrode placed in the epidural space through the upper lumbar ($L_2$-$L_3$) section and referenced to the adjacent paraspinal muscle with a platinum needle electrode (Grass-Astromed, USA). The analog signals were amplified and filtered (bandpass 30–3000 Hz) and 32 single sweeps were averaged on an analog/digital converter (Axon-Instruments, USA). The MEPs were stored in a hard disk of a computer for subsequent offline analysis.

We determined the stereotaxic coordinates of the cerebral cortex in which the stimuli produced the best responses. For all rats, which corresponded to 2.04 mm in AP coordinates, following Paxinos and Watson atlas [21] that overlapped with previous studies [23, 24]. The amplitude of MEPs was quantified as the voltage difference from the positive to the following maximum negative peak.

At the end of the experiments, the rats were sacrificed with an overdose of sodium pentobarbital 70 mg/Kg (Sedalmerk, México), then a craniotomy was made using gouge and shears to extract the brain. The area of cortical stimulation was identified using a Nikon stereo microscope (Japan) coupled to a digital camera.

## Procedures and apparatus

For the novelty-induced locomotor test, we used a transparent acrylic cage (46 x 32 x 20 cm) with sterilized wood shavings on the floor with an incandescent lamp atop (149 lux) following the criteria [25]. After 15 minutes of habituation, each animal was put alone in the acrylic cage

and observed immediately after an intraperitoneal injection of 4-AP (1 mg/Kg) or an equivalent volume of physiological saline solution as a control group. Two observers, one of them blinded about the condition, recorded the number of horizontal locomotion (ambulation bouts), vertical displacements (rearing plus wall leanings bouts), and tremor bouts. After each observation, the acrylic cage was thoroughly cleaned with soap and water.

### Drugs

4-aminopyridine was provided by Sigma-Aldrich (St. Louis, MO, USA). The drug was dissolved in physiological saline solution before usage, then put in a Vortex until obtained a homogeneous solution. 4-aminopyridine and the volume was adjusted to 1 mL/Kg which is equivalent to a final concentration of around 250 mM when diluted in the body below seizure activity [26]. All animals first received a control injection of physiological saline solution to have basal SSEPs and then 4-AP administration and the subjects recorded 90 min after injection.

### Human patient with H-ABC leukodystrophy

Somatosensory evoked potentials were recorded from median and tibial nerves with a Nicolet Viking Quest system (Madison, WI, USA) in a patient suffering from H-ABC and in a healthy control subject for comparisons. The stimulus electrodes were placed at the skin, 2 cm proximal to the wrist, and at the medial aspect of the ankle, respectively. Monophasic pulses of 0.2 milliseconds of 12.5 mA CC were used as a stimulus at a rate of 4 Hz. The recordings were bandpass filtered from 10–3000 Hz and peak-to-peak amplitudes and latencies were measured. All procedures followed ethical guidelines and we had an informed consent that was signed by the parents.

### Tissue preparation and image acquisition

The protocol used for tissue preparation was described previously [14]. Briefly, three *taiep* and three WT rats aged three months were anesthetized with a mixture of ketamine/xylazine (75 mg/Kg and 5 mg/Kg, respectively), and their tissues fixed by transcardial perfusion with 4% formaldehyde in phosphate saline solution (PBS). Spinal cords were removed by laminectomy and divided into regions (cervical, thoracic, and lumbar). The lumbar regions were postfixed with formaldehyde solution (4% in PBS) for 60 minutes at room temperature, immersed in sucrose solution (30% in PBS) at 4˚ C for 24 h, and frozen using tissue freezing medium (Leica, USA). Forty-micron slices were obtained in a CM 1860 cryostat (Leica, USA). Sections were marked with an anti-neurofilaments-200 antibody (N4142, Sigma, USA) and immunostained with Alexa Fluor 488 (Thermo Fisher Scientific, USA). Myelin sheaths were stained with Fluoromyelin red (Thermo Fisher Scientific, USA) as indicated in the datasheet. Nuclei were stained with DAPI (Thermo Fisher Scientific, USA). The samples were observed with a LSM 710 Zeiss confocal microscope (Germany), using a 25x oil-immersion objective. Identical acquisition settings were used for *taiep* and WT control images using FIJI software to reconstruct and analyze the fluorescent images [27].

### Statistical analysis

Statistical differences were analyzed with appropriate tests, indicated in figure legends, comparing the WT and *taiep* groups. For all experiments, P< 0.05 was considered significant. The effects of 4-AP were analyzed with paired Student t-tests using Sigma-Plot v. 11.0 software (Systat Software Inc., CA, USA). For the fluorescence intensity averaged values were obtained

from six different regions of interest in the ventral and dorsal funiculi, respectively. Statistical analyses were carried out with GraphPad (version 9.1.0 GraphPad Software Inc., La Jolla, CA, USA). Data are plotted as mean ± SEM of three WT and three *taiep* rats.

## Results

### Somatosensory evoked potentials in *taiep* rats

In eight adult WT rats, two negative peaks were obtained in the surface of the spinal cord when the posterior tibial nerve was stimulated with squared pulses of supramaximal intensity (10xT) of 100 μsec duration. The SSEPs changed their morphology and amplitude depending on the spinal cord level at which the traces were recorded. The maximum amplitude was obtained between the $T_{12}$ and $L_6$ spinal cord levels and the potentials were characterized by a single negative peak with an upward displacement in WT rats (see Fig 1A, black arrow).

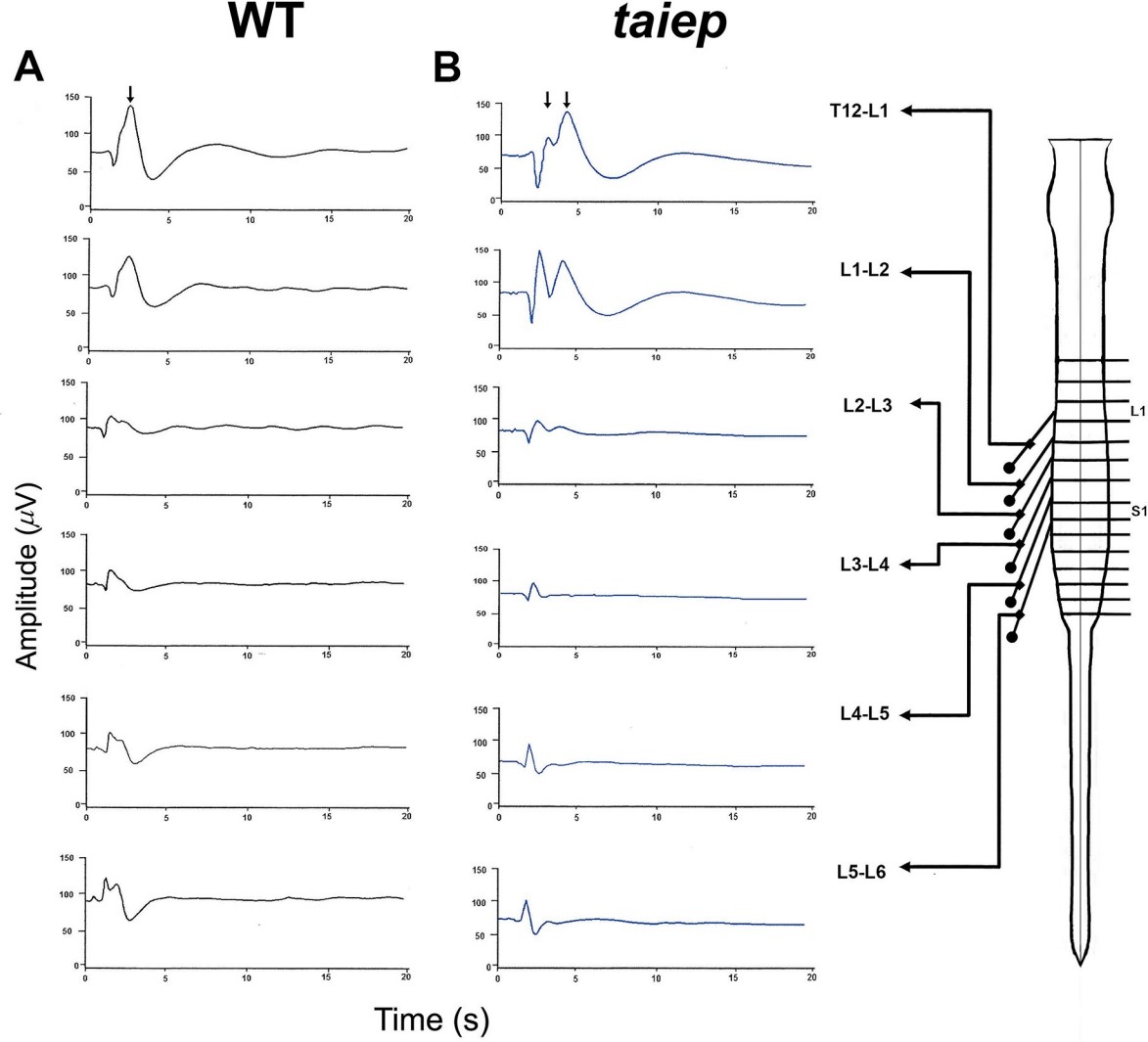

**Fig 1. Somatosensory evoked potentials produced by peroneal nerve stimulation at different spinal cord levels. A)** In wild type Sprague-Dawley (WT) male rats the somatosensory evoked potentials have a characteristic negative wave (upward deflection, black arrow) followed by a single positive (downward deflection), being maximum at $T_{12}$-$L_1$ and $L_2$-$L_1$ interspinal levels. **B)** Male *taiep* rats have two negative peaks (upward deflections, black arrows) being the second ($N_2$) clearly delayed followed by positive evoked wave that is similar to that obtained in WT rats.

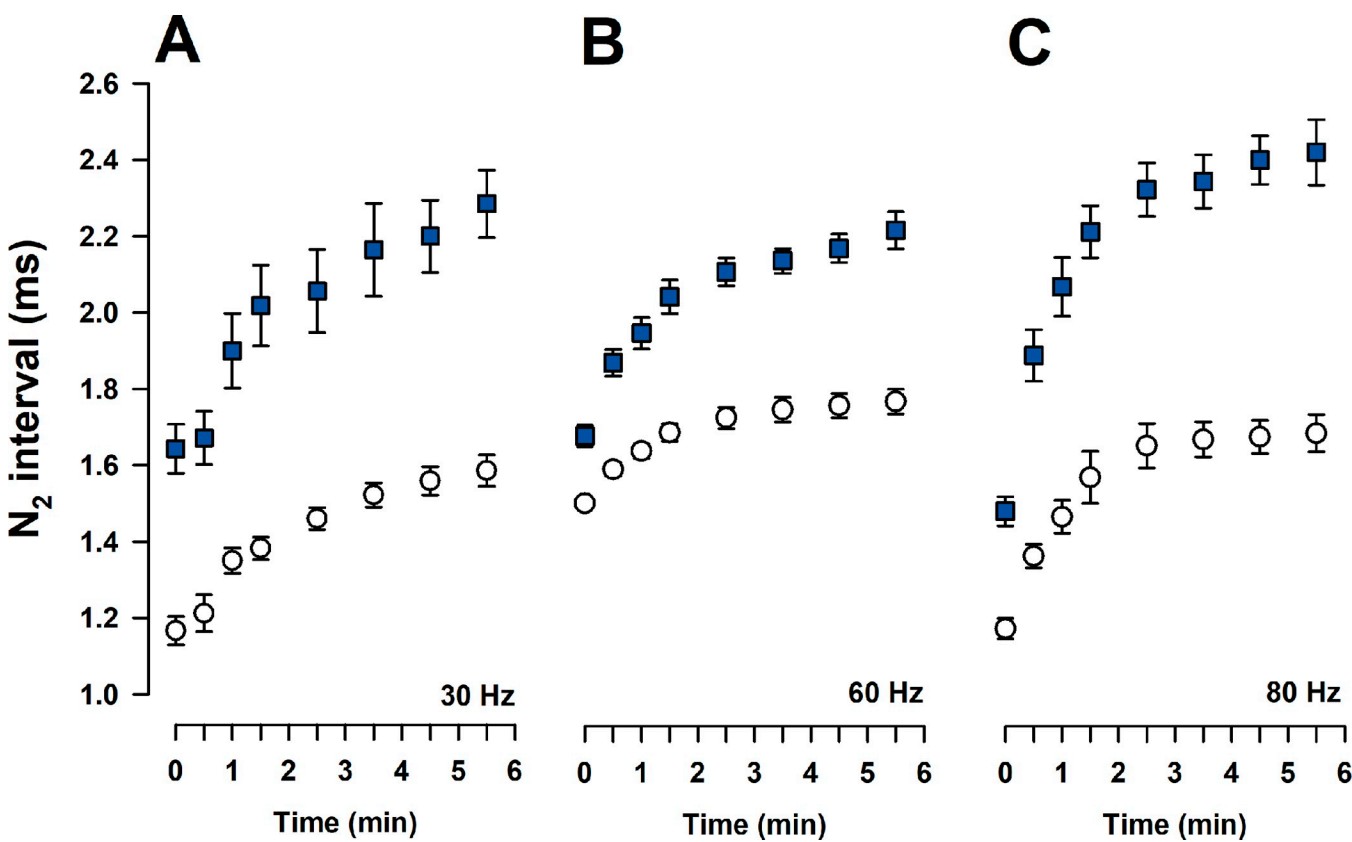

**Fig 2. Somatosensory evoked potentials in *taiep* and wild type Sprague-Dawley male rats. A)** Iterative stimulation on the peroneal nerve at 30 Hz produced a progressive delayed $N_2$ wave of the evoked potentials in *taiep* from 1.65 to 2.31 msec (blue squares) with respect to WT male rats (1.55 msec, empty circles). **B)** The iterative stimulation with 60 Hz produced a rapid increase in the intervals among successive $N_2$ evoked potential in *taiep* rats (2.15 msec), but it is not the case in WT because they have a stable $N_2$ evoked potential around 1.72 msec. **C)** The increase in successive $N_2$ evoked potential was faster in *taiep* with 2.33 msec intervals with respect to WT with 1.65 msec intervals when peroneal nerve stimulation is 80 Hz. The data is the mean ± E.E.M. of eight subjects.

In eight *taiep* rats, we recorded two clearly negative components due to a second negative ($N_2$, second arrow) delayed central response (see Fig 1B, black arrows). The maximum separation between these two peaks was recorded at the $T_{12}$ and $L_1$ and $L_2$-$L_1$ interspinal levels which decreased in upper lumbar spinal cord levels. The SSEPs clearly reflect asynchrony in the central components of the evoked potentials due to demyelination.

Considering that SSEPs are the sum of the electrical fields produced by action potentials plus the postsynaptic potentials that produced a second negative peak $N_2$, we analyzed the temporal course of successive $N_2$ evoked potentials in WT and *taiep* rats using repetitive simulation in the posterior tibial nerve with different frequency rates. Our results showed that successive $N_2$ evoked potentials were delayed in *taiep* rats (see Fig 2, blue squares) with respect to WT rats that are capable to follow the different frequencies of stimulation tested (empty circles).

In fact, $N_2$ peak of *taiep* SSEPs were delayed, and they were highly dependent on the frequency of stimulation, because with 30 Hz there was a progressive delay in the successive evoked potentials to reach 2.31 msec. When stimulation increased to 60 Hz, the SSEPs reached a plateau at 1.72 msec interval among successive $N_2$ peaks in WT compared with a mean of 2.15 msec in *taiep* rats (see Fig 2B). Stimulating with 80 Hz, in the WT rats they had a stable interval in the evoked potentials, but in *taiep* rats, there were delayed among $N_2$ successive

peaks to reach 2.33 msec in successive intervals (see Fig 2C). These results imply that demyelination change the recovery cycle of the action potentials and probably a change in the safety factor for the central transmission of action potentials in the intraspinal collaterals of the stimulated nerve.

## Somatosensory evoked potentials after 4-aminopyridine administration in wild type and *taiep* rats

SSEPs were evaluated in eight adult WT and eight *taiep* rats after 90 min of intravenous (i.v.) administration of 1 mg/Kg of 4-AP. In WT rats 90 min after i.v. of 4-AP the amplitude of $N_2$ evoked response increased in their amplitude (Fig 3A), the $N_2$ amplitude of the evoked response significantly increased after 4-AP with respect to control SSEPs (dark bar; U = 62.5; $P<0.05$, see Fig 3B).

In *taiep* rats 15 min after injection, the $N_2$ somatosensory evoked potential showed a significant increase in their amplitudes with a 13.2% (see Fig 3C, blue dotted line), which reached a maximum response of 57% ninety min after 4-AP, with shorter latencies from 5.36 ± 0.03 sec in control $N_2$ evoked potential, to just 5.06 ± 0.02 sec dotted SSEPs after 4-AP administration (blue dotted line; Fig 3C). The maximum responses were obtained at 90 min interval with a latency of 4.93 ± 0.03 sec. In Fig 3D a significant increase in the amplitude of $N_2$ evoked potentials after 90 min of 4-AP injection in *taiep* rats (blue bar; $t_{(29)}$ = 4.3; $P<0.01$) with respect to SSEPs obtained after physiological saline administration.

To further characterize the behavioral effects of 4-AP systemic injection, we also evaluated the effects of this drug on the number of ambulation and tremor bouts with respect to saline-treated animals. Our results showed a significant increase in horizontal ambulation ($t_{(29)}$ = 3.6; $P<0.05$, see Fig 4), but not in rearing (vertical displacements) or in tremor bouts. Our results clearly showed that 4-AP is a potential therapeutic agent for the treatment in patients with H-ABC or with other leukoencephalopathies.

## Motor evoked potentials in *taiep* rats

Cortical stimulation in the hindlimb area (Fig 5A shaded area) induced motor evoked potentials (MEPs) in the lumbar spinal cord in WT rats, with an initial deflection due to direct (D) activation of the pyramidal tract followed by two indirect (I-1, I-2) waves generated by the reactivation of the pyramidal tract (see Fig 5B). In *taiep* rats, cerebral cortex stimulation produced a delayed MEP with fused direct and indirect waves (see Fig 5C).

At 30 Hz stimulation, MEPs amplitudes decreased and were blocked in *taiep* rats (see Fig 6A, blue squares), while WT are capable to follow successive MEPs with the different stimulation frequencies tested from 5 to 30 Hz (see Fig 6A, empty circles). Importantly, the corticospinal conduction velocities significantly decreased in *taiep* rats with just 13.7 ± 0.6 m/s (blue bar) with respect to 42.9 ± 1.2 m/s in WT rats (empty bar; $t_{(12)}$ = 42, $P<0.05$; see Fig 6B). These results showed that this efferent pathway is affected in an equivalent way as the central components of the SSEPs supporting that evoked potentials could be useful tool in clinical neurophysiological evaluations.

## Somatosensory evoked potentials in a human patient with H-ABC

Median nerve stimulation showed clear SSEPs with N19 latency differences between left and right nerves of 0.20 msec and P22 of 1.80 msec, respectively, and delayed compared with the healthy patient (black traces). The amplitude difference between the left and right nerve was -0.04 μV. However, tibial nerve stimulation in the left and right hindlimbs did not induce any SSEPs in H-ABC patient (Fig 7 blue traces), but normal SSEPs were obtained in healthy

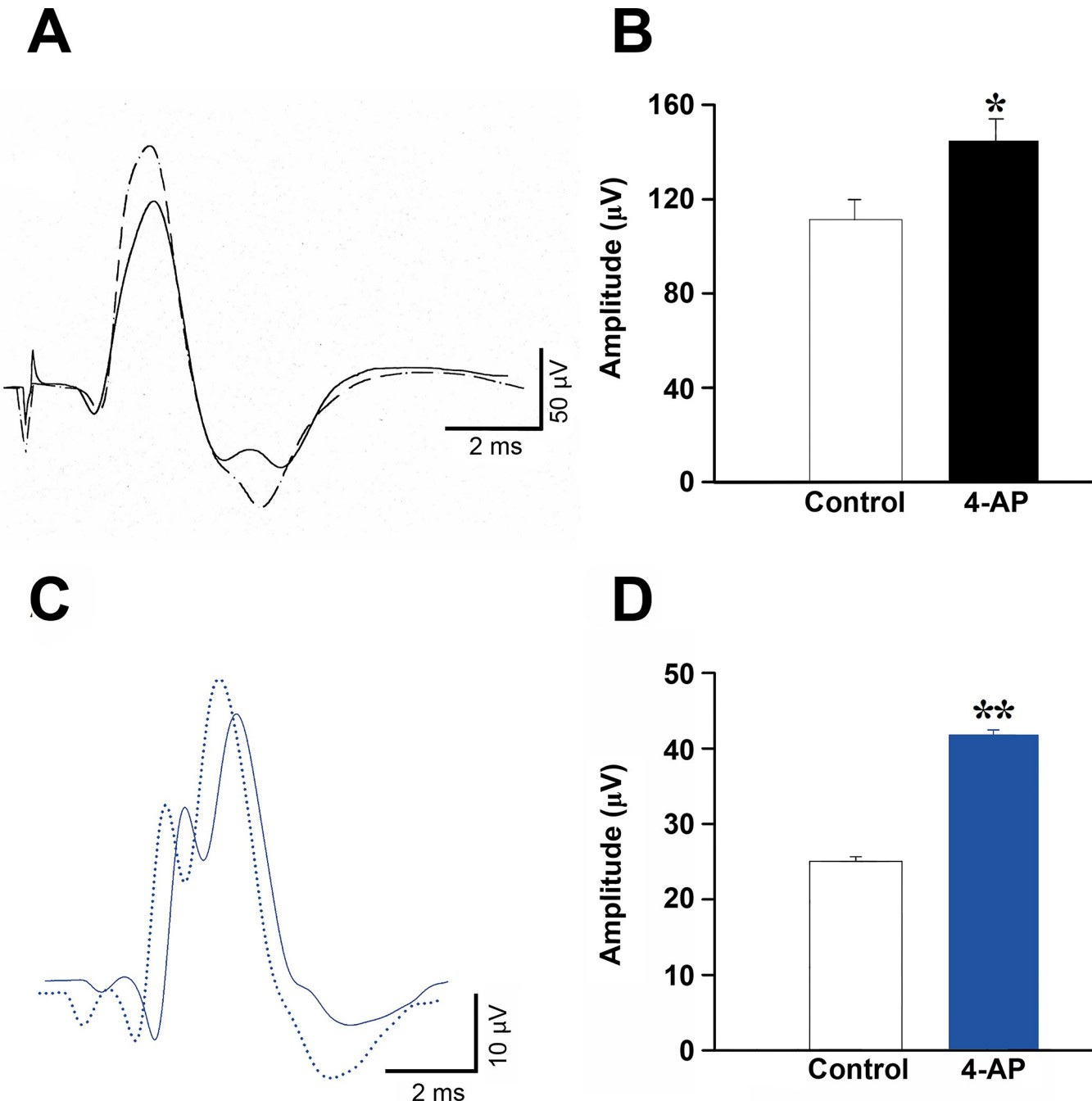

**Fig 3. The systemic administration of 4-aminopyridine increased the somatosensory evoked potentials in *taiep* rats. A)** The somatosensory evoked $N_2$ evoked potential increased after systemic administration of 4-aminopyridine (4-AP, dotted lines) with respect to control evoked potentials (continuous lines) in WT male rats with a shorter latency. **B)** The somatosensory evoked potentials significantly increase after intravenous administration of 1 mg/Kg of 4-AP (U = 62; $P < 0.05$, Mann-Whitney U test) with respect to saline-treated rats (empty bar). **C)** The somatosensory $N_2$ evoked potential increased after systemic administration of 4-aminopyridine (4-AP, blue dotted lines) with respect to control evoked potentials (blue continuous lines) in *taiep* rats. **D)** The $N_2$ evoked response significantly increased its amplitude and reduced its latency after the administration of this potassium channel blocker (blue bar; t $_{(29)}$ = 4.3, $P < 0.001$) with respect to control conditions. The data is the mean ± E.E.M. of eight subjects.

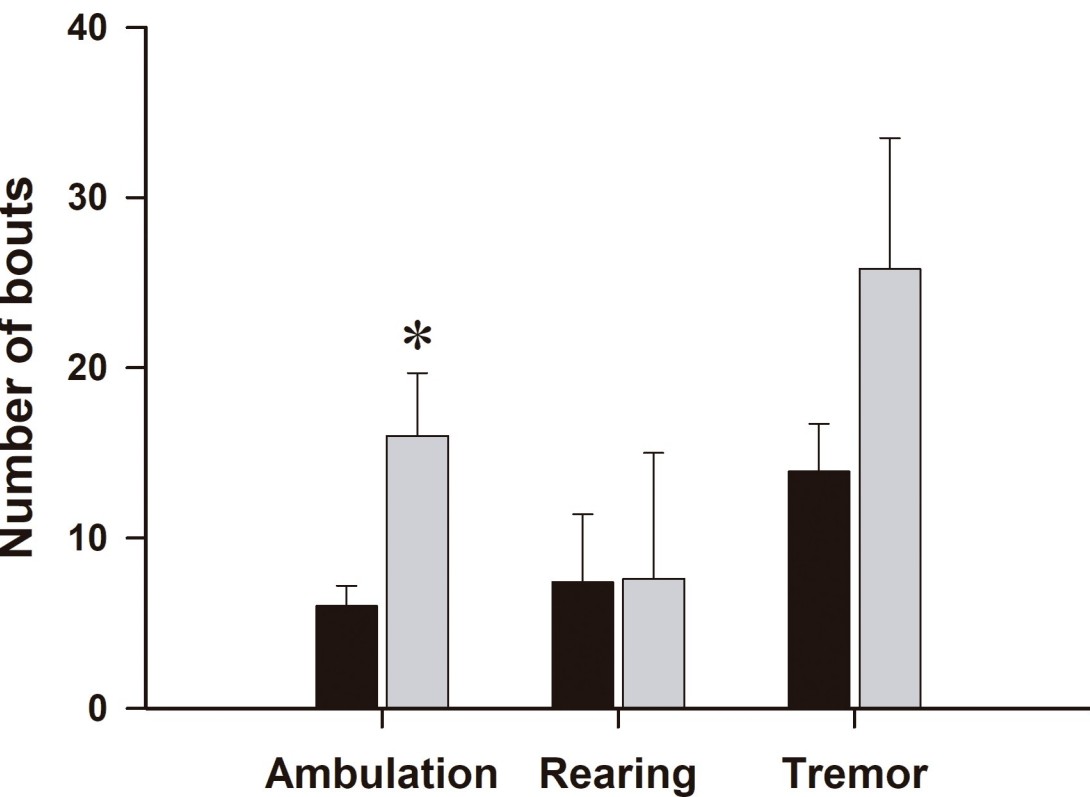

**Fig 4. Number of ambulation, rearing, and tremor bouts in control and after 4-aminopyridine administration in male *taiep* rats.** In the novelty-induced locomotor test, 4-aminopyridine (4-AP) i.p. injection of 1 mg/Kg significantly increased the ambulation bouts ($t_{(29)} = 3.6$; $P< 0.05$), but not rearing and tremor bouts. The data is the mean ± E.E.M. of eight subjects.

patient, and they had shorter latencies due to normal myelination of the central pathways (Fig 7 black traces). Our results showed that in the animal model and a patient with H-ABC neurophysiological evaluation is a powerful technique to determine the amount of myelin loss and its changes along the evolution of the disease.

## Morphological characteristics of the spinal cord in *taiep* and wild type Sprague-Dawley rats

Immunostaining of the dorsal and ventral funiculi in the spinal cord of WT rats revealed the conventional appearance of myelin staining (red) and neurofilaments (green) in the dorsal column of WT (Fig 8A) and a decrease in the myelin staining in *taiep* rats (Fig 8B). Note the images of individual channels show bellow the spinal cord slices allow full appreciation of the severe demyelination and the full display of their filaments in severe demyelinated fibers in green. The myelin fluorescence intensities in the dorsal columns of *taiep* were significantly lower with respect to WT male rats (Fig 8E; $P< 0.0001$, Mann-Whitney U test). A similar pattern of lower myelin staining was obtained in the ventral funiculus in WT (Fig 8C) with respect to that obtained in the ventral part of the spinal cord in *taiep* rats (Fig 8D). The myelin fluorescence intensity was significantly lower in *taiep* with respect to WT rats (Fig 8F; $P< 0.0001$, Mann-Whitney U test). These results showed that dorsal and ventral spinal cord white matter were affected in *taiep* rats and correlated with the previous neurophysiological data.

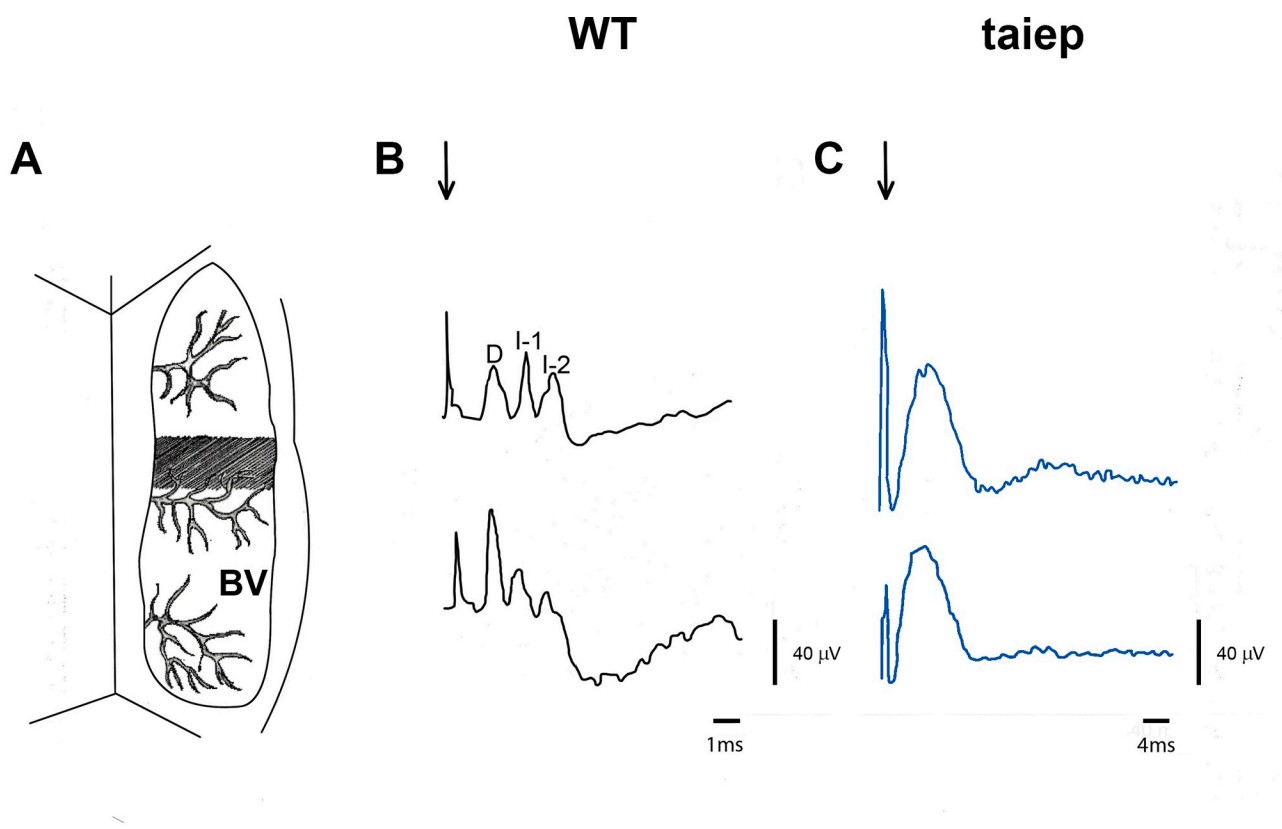

**Fig 5. Motor evoked responses differ in *taiep* and wild type Sprague-Dawley male rats. A)** The grey area is the hindlimb area of stimulation which produce maximum motor-evoked potentials in lumbar spinal cord. **B)** In Sprague-Dawley (WT) male rats have a direct wave (D) followed by two indirect waves (I-1 and I-2) due to reverberating activity in the pyramidal pathway. Downward arrow is the stimulus artifact. **C)** In *taiep* rats, direct and indirect waves are fused, which produced a characteristic motor-evoked response in this tubulinopathy rat model. The data is the mean ± E.E.M. of eight subjects.

## Discussion

Recently, our group showed that the *taiep* rat carries a point mutation in the tubulin β 4A gene [7]. Moreover, this rat displays histopathological and magnetic resonance imaging similar to those observed in patients suffering from H-ABC, including hypomyelination in different central pathways and atrophy of the basal ganglia and cerebellum, making this spontaneous mutant rat an ideal model for studying tubulinopathies [7, 8, 14]. Furthermore, our research group demonstrated that the central components of the auditory evoked potentials in *taiep* rats are significantly altered in their morphology and with delayed central components that are due to the demyelination process with a similar response in a Mexican patient suffering from H-ABC [8]. These results support that, besides the structural damages in TUBB4A and the corresponding hypomyelination of the central white matter, the clinical neurophysiology evaluation can be a valuable tool to determine the integrity of myelin sheaths in human patients suffering from this tubulinopathy and it could be useful in other leukodystrophies or leukoencephalopathies [28, 29]. Consequently, the availability of the *taiep* rat allows the evaluation of other neurophysiological evoked responses such as SSEPs and MEPs and compare them with wild type rats, we also compare healthy children with respect to those affected of H-ABC.

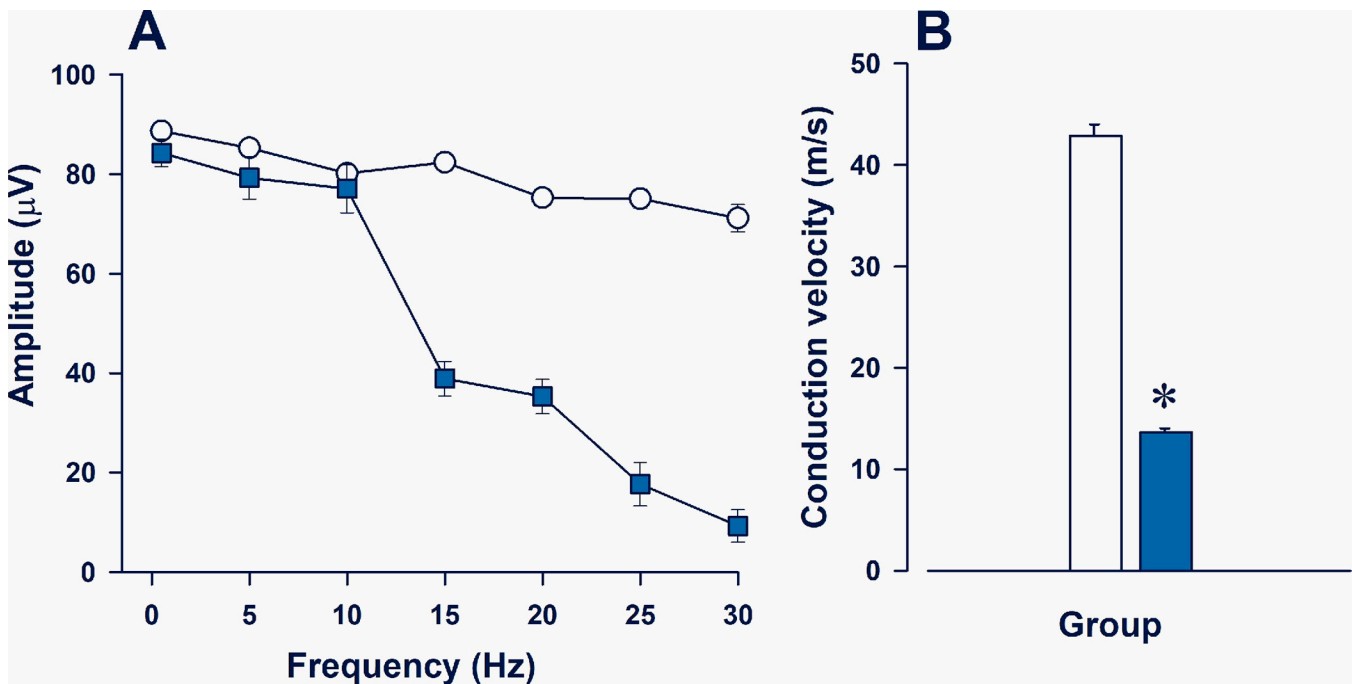

**Fig 6. The corticospinal pathway is more susceptible to iterative stimulation and have lower conduction velocity in *taiep* rats. A)** Motor-evoked potentials (MEPs) induced by iterative electrical stimulation on the cerebral cortex showed a decrease in their amplitudes when increasing the stimulation rate from 5 to 30 Hz in *taiep* (blue squares), but not in WT rats (empty circles). **B)** The conduction velocity of the corticospinal tract significantly decreased in *taiep* to 13.7 m/s (blue bar) with respect to WT with 42.9 m/s (empty bar; t $_{(12)}$, = 42; **$P < 0.05$). The data is the mean ± E.E.M. of eight subjects of each group of rats.

In this work, we showed that motor-evoked potentials in the wildtypehave a characteristic early response due to the direct stimulation of the pyramidal tract (D-wave) followed by several indirect waves (I1 and I2) due to the reverberating activation of the corticospinal pathway [30–34]. Adult *taiep* rats have a fused MEPs (see Fig 5), which could be due to the demyelination process that impairs the synchronicity of the action potentials. In fact, in the neonatal spinal cord and in the hippocampal *in vitro* preparations, *taiep* rats had an asynchronous transmission of the action potentials and in their corresponding postsynaptic currents [35, 36]. Importantly, the mean conduction velocity of the pyramidal tract in wildtype rats is 42.9 ± 1.2 m/s, which is similar to previous reports [31, 37]. Additionally, our results showed that in WT male rats followed 30 Hz stimulation without remarkable changes in the amplitude of MEPs, similar to what was obtained by Gorman [38]. Nevertheless, *taiep* rats have single MEPs with a significant reduction in the conduction velocity of the corticospinal pathway with only 13.7 ± 0.6 m/s which is just one-third of the normal conduction of this descending pathway. The severe decrease in the conduction velocity of the pyramidal tract pathway correlated with a decrease in the myelination of the corticospinal tract measured through detailed electronic microscopy measurements including g ratios [17].

These neurophysiological responses can be explained as the result of the reduction in the myelin thickness in this descending motor tract [17]. Comparable results were found in patients suffering from Pelizaeus-Merzbacher disease (PMD) when the corticospinal tract was stimulated using magnetic fields [2, 39]. On the other hand, we found that SSEPs are delayed and with different morphology in adult *taiep* rats with respect to WT. It is relevant, that clinical neurophysiological studies on patients with PMD, adrenoleukodystrophy (ALD), or metachromatic leukodystrophy (MLD) SSEPs display longer latencies and significant changes in the intervals of the central evoked responses equal to that obtained in our patient with H-ABC

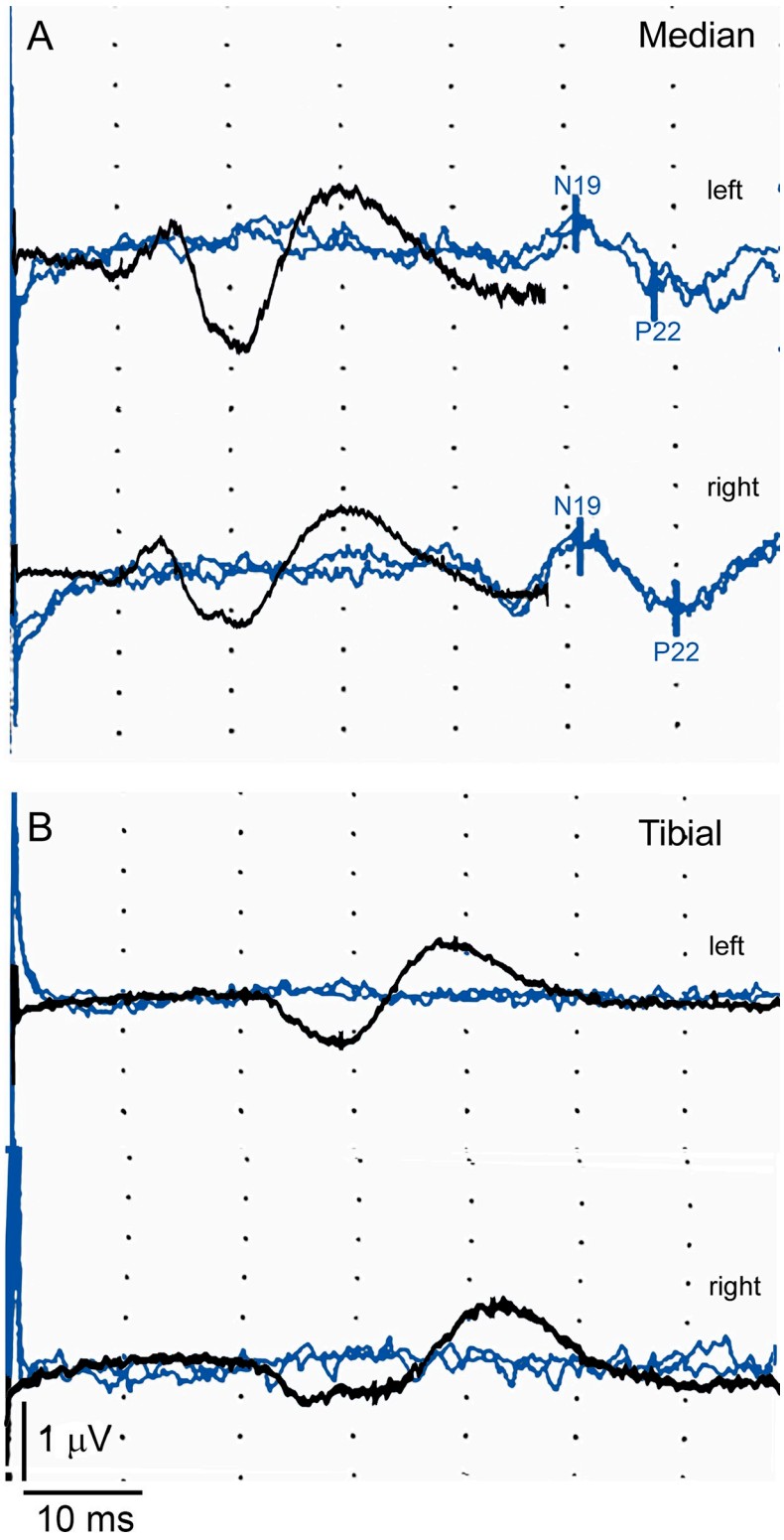

**Fig 7. Defects in the conduction of somatosensory evoked potentials of median and tibial nerves in a healthy child and in H-ABC patient. A)** Somatosensory evoked potentials of left and right median nerves showing an N19 latency at the cortex with 50.9 and 51.1 msec delay, respectively, and P22 latencies with 58.0 and 59.6 msec, respectively (blue traces). **B)** Tibial nerves did not evoke any potentials in both hindlimb nerves (blue traces) with respect to healthy children (dark traces). Recordings were made at a rate of 4 Hz, with a stimulus intensity of 12.5 mA with a square pulse

duration of 2 msec. The SSEPs from a healthy patient, recorded with the same equipment are shown in black traces for comparison.

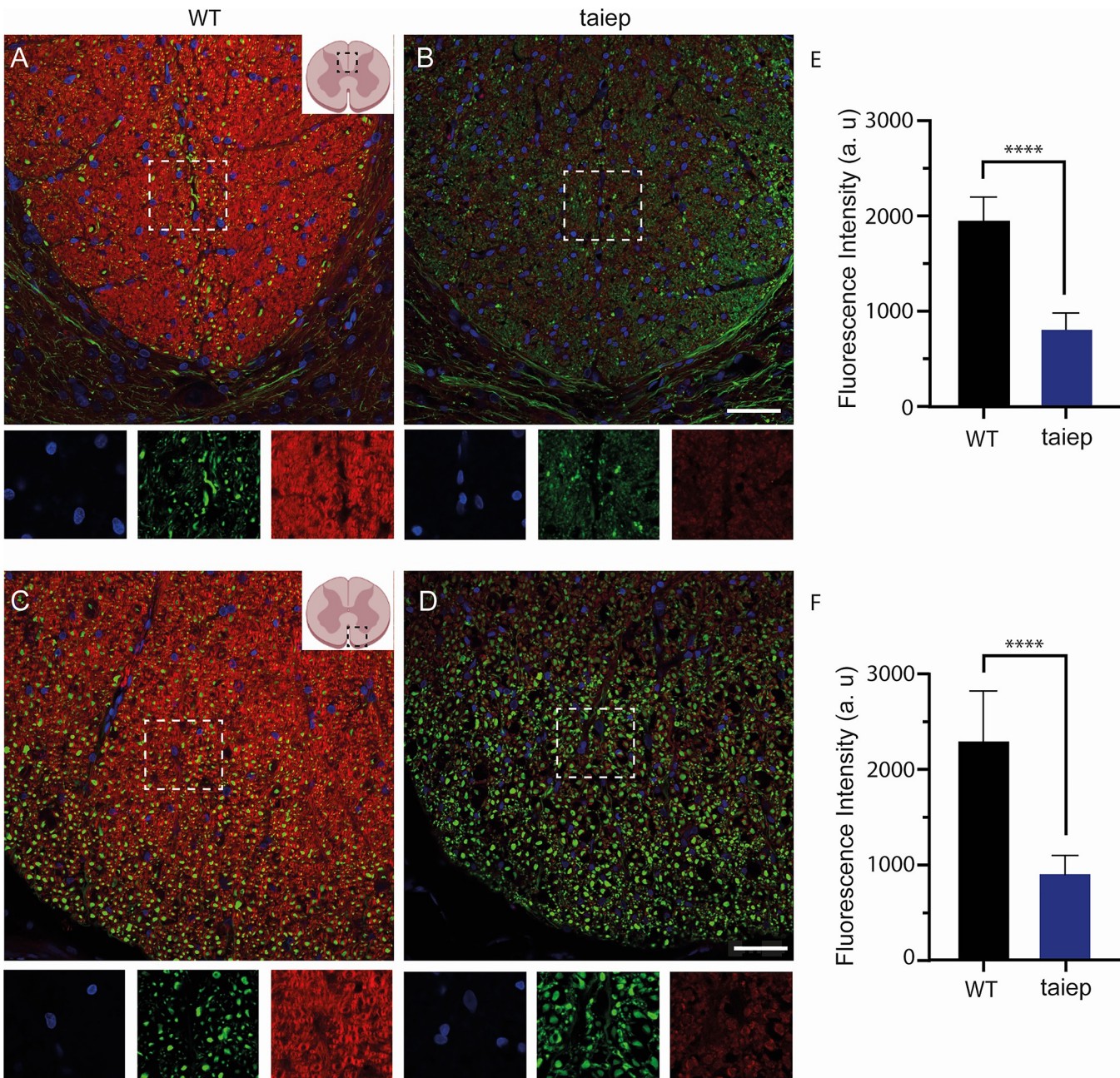

**Fig 8. Hypomyelination and demyelination in the spinal cord of *taiep* rats.** Representative confocal images of transversal sections of spinal cords at the lumbar level in the ventral and dorsal funiculi (as shown in the drawings) of wild type Sprague-Dawley (WT; panels **A** and **C**) and *taiep* rats (panels **B** and **D**). The myelin staining (red) is even throughout the section, while neurofilament staining (green) shows a spotty appearance. In the *taiep* rat, the myelin staining is drastically reduced (**B** and **D**). **E)** In the dorsal column a significant decrease in red the myelin fluorescence intensity in *taiep* rats (blue bar; $P < 0.0001$, Mann-Whitney U test) with respect to WT rats (black bar). **F)** In the ventral funiculus there is also a significant reduction of red myelin fluorescence intensity in *taiep* (blue bar) with respect to WT male rats (black bar; $P < 0.0001$, Mann-Whitney U test). Nerve fibers devoid of myelin sheaths fully display their NF content (green). Images of individual channels corresponding to the white dotted squares are shown below each image and allow us to appreciate the severe demyelination in the dorsal and ventral funiculi. Green: neurofilaments, red: myelin, blue: DAPI. Scale bar 50 μm. Myelin fluorescence levels were measured in 6 regions of interest. The data is the mean ± S.E.M. of six subjects of WT and *taiep* rats.

[40, 41], supporting that evoked potentials are a useful tool to estimate the amount of lesion of the white matter in leukodystrophies or even in other leukoencephalopathies with different genetic backgrounds [12, 28, 42, 43].

It is important to emphasize that SSEPs in *taiep* rats have a delayed central component and are not able to follow repetitive stimulation. On the other hand, intravenous administration of 4-AP, a well-known blocker of potassium channels, increases the amplitude of SSEPs and reduce their latencies that are similar to data obtained in patients with spinal cord transections to restore axonal transmission of action potentials [44]. Of relevance in this context is our previous report that 4-AP induced a significant increase in the amplitudes and broadening of the compound action potentials in the *in vitro* optic nerve preparation [18]. Here we demonstrated that 4-AP increases ambulation episodes in *taiep* rats, but not vertical displacements or number of tremor bouts (see Fig 4). These effects could be due to the well-known effect to increase the release of neurotransmitters in the cat spinal cord [45]. In the same way, in demyelinated axons, it has been demonstrated that 4-AP increases neurotransmitters release [46]. Additionally, in demyelinated axons, 4-AP partially restores the transmission of the action potentials due to augmented synaptic efficacy between spindle afferent fibers and moto-neurons in the cat spinal cord [47]. All these effects can underlie the increase in the $N_2$ response of the SSEPs after 4-AP administration in *taiep* rats and in the ambulatory capabilities when exposed to a new environment (Figs 3 and 4), which is similar to what obtained in rats and human patients with spinal cord lesions in which 4-AP enhanced MEPs [44, 48, 49].

Finally, 4-AP was shown to partially restore visual and locomotor responses in multiple sclerosis patients [50, 51], also partially improving sensory and motor deficits [52]. It has been demonstrated that sustained released 4-aminopyridine using Fampiridine-SR orally administered with different doses twice daily in patients with chronic incomplete spinal cord injury improved motor performance, being the drug iwell tolerated and with minimum adverse events [53]. In fact, this long-acting formulation with 10 mg dose every 12 hours in multiple sclerosis patients clearly improve gait parameters [54], being an adequate pharmacological tool for the treatment of different demyelinating diseases, because it is capable to increase synapse efficiency and improving the transmission of action potentials through demyelinated axons. It is relevant that the 4-AP dose assessed in the present experiments is below the threshold of seizure activity in mice [55] or in the hippocampal *in vitro* preparation [26] and could be the base for future clinical evaluations.

We focused on the neurophysiological evaluation using SSEPs and MEPs in adult male *taiep* rats because somatosensory evoked potentials are a reliable way to evaluate the conduction of action potentials from the periphery through primary afferent synapses with second-order interneurons which produced a characteristic field potential [56]. Our data could be a frame for clinical studies in which an integral evaluation of evoked potentials produced by different sensory modalities such as sensorial, auditory, or visual potentials must be evaluated to estimate the progression of myelin diseases and measure the effects of different therapeutical strategies.

It is necessary for a thorough neurological examination, MRI imaging studies and a detailed clinical neurophysiological evaluation in patients diagnosticated with H-ABC or with other leukoencephalopathies and should be mandatory in all patients to obtain clinical neurophysiological studies at primary care facilities because are a feasible option, with low-cost and can be applied several times because is a safe technique and allow to evaluate conduction velocities and the transmission of action potentials through central pathways or as a way to evaluate the effect of new therapies.

It is quite remarkable that in our H-ABC patient, we were able to demonstrate a clear difference in the forelimb and hindlimb SSEPs which differ with respect to that recorded in healthy

**Fig 9. *Taiep* rats meet criteria as an adequate model of H-ABC.** *Taiep* rats is an adequate model of the human leukodystrophy H-ABC because they have face validity in base of similar MRI findings, they had a point mutation in tubulin β 4A gene that altered structurally the corresponding protein (TUBB4A). Additionally, the construct validity is supported by similar somatosensory and motor-evoked potentials and with their behavioral responses in a novelty-induced locomotion measured through ambulation bouts.

children that have shorter latencies. It is here where the correlation with the animal model becomes crucial as pathology material from patients is always unavailable. The fasciculus gracilis and corticospinal tracts in *taiep* rats, which contain mainly small-caliber axons, and the ventral tracts which have mainly large-caliber axons are both affected (Fig 8). Additionally, SSEPs and MEPs could be used as the main clinical tool to determine the effects of different new therapies including genetic or immunological manipulations.

Recently, we showed that the long-term administration of 50 mg/L of taurine (2-aminoethanesulfonic acid) in drinking water from gestational age 15 to postnatal day 240 decreases nitrites and lipoperoxidation in the central nervous system in adult male *taiep* rats and this amino acid increases cell proliferation in the dentate gyrus and in the pons in our tubulinopathy rat model [57]. Taurine improves innate motor behaviors such as cliff aversion, grip strength, and forelimb suspension with respect to untreated *taiep* rats. Taurine is also capable of significantly decreasing the number of immobility episodes induced by gripping the tail of adult *taiep* rats [57]. These results also show that early intervention with taurine could change the progression of myelination in *taiep* rats. These results with taurine and the already reported effects of 4-AP could be the frame to evaluate in patients with H-ABC or other leukoencephalopaties.

Notably, there are only two available mouse models of H-ABC. However, both have short survival times, i.e., the homozygous tubb4a$^{D249/D249}$ has swallowing troubles and righting reflex at 35–40 PND age at which they were sacrificed [58], while the Jittering (Jit) mouse has a degenerative phenotype affecting cerebellar granule cell neurons and myelination, and it typically survives up to 50 PND [59]. Clearly, these models do not resemble human patients'

clinical features; on the other hand, they do not allow developmental electrophysiological studies or different therapeutic tests as it can be done with *taiep* rats that have a normal life expectancy [60].

## Conclusions

*Taiep* rat is the most reliable animal model for studying H-ABC because it has a face validity of the main pyramidal and extrapyramidal motor signs of this leukodystrophy, they had characteristic imaging features in the white matter, basal ganglia, and cerebellum using MRI analysis and they had a point mutation in the tubulin β 4a gene and its corresponding TUBB4A protein (Fig 9, left panel).

Therefore, a wide variety of different approaches can be used to study the disease, among those the analysis of different sensory or motor evoked potentials, biophotonics of the central white matter, and evaluation of some therapeutic approaches (Fig 9, right panel) [61, 62]. Due to the normal life expectancy of this model, long-term evaluation of the effects of cellular and genetic manipulations is feasible, opening the possibility to carry out experiments from the bench to the bedside with a possible predictive validity for testing new therapeutic options with long-term duration, as we already demonstrated with taurine administration during pregnancy (in *taiep* dams, Fig 9 lower panel), or even short-term effects with 4-AP in the present results. In conclusion, electrophysiological evaluation of sensory and motor evoked potentials is a useful tool to determine the progression of leukodystrophies and to validate the effects of new therapeutical options. Additionally, our results support the use of 4-aminopyridine and its derivatives as a possible additional treatment for H-ABC patients in the near future.

## Acknowledgments

Thanks to Dr. Yonatan Puón and Dra. Norma Chantal-Seoane for editing the English-language text.

## Author Contributions

**Conceptualization:** Jose R. Eguibar, Carmen Cortes, Victor H. Hernandez.

**Data curation:** Jose R. Eguibar, Carmen Cortes, Alejandra Lopez-Juarez.

**Formal analysis:** Carmen Cortes, Victor H. Hernandez, Alejandra Lopez-Juarez, Valeria Piazza, Diego Carmona.

**Funding acquisition:** Jose R. Eguibar, Carmen Cortes, Victor H. Hernandez, Valeria Piazza.

**Investigation:** Jose R. Eguibar, Carmen Cortes, Victor H. Hernandez, Alejandra Lopez-Juarez, Diego Carmona, Anke Kleinert-Altamirano, Blanca Morales-Campos, Emilio Salceda, Manuel Roncagliolo.

**Methodology:** Carmen Cortes, Victor H. Hernandez, Alejandra Lopez-Juarez, Valeria Piazza.

**Project administration:** Jose R. Eguibar, Carmen Cortes.

**Resources:** Jose R. Eguibar.

**Supervision:** Jose R. Eguibar, Carmen Cortes, Victor H. Hernandez, Valeria Piazza.

**Validation:** Jose R. Eguibar, Carmen Cortes, Victor H. Hernandez.

**Writing – original draft:** Jose R. Eguibar, Carmen Cortes.

**Writing – review & editing:** Jose R. Eguibar.

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
