## [Decision Letter · Decision Letter 0]

29 Sep 2023

PONE-D-23-189404-aminopyridine improves evoked potentials and ambulation in the taiep rat: a model of hypomyelination with atrophy of basal ganglia and cerebellum.PLOS ONE

Dear Dr. Eguibar,

Thank you for submitting your manuscript to PLOS ONE. After careful consideration, we feel that it has merit but does not fully meet PLOS ONE’s publication criteria as it currently stands. Therefore, we invite you to submit a revised version of the manuscript that addresses the points raised during the review process.

We look forward to receiving your revised manuscript.

Kind regards,

Nejat Mahdieh

Academic Editor

PLOS ONE

Journal Requirements:

"This work was supported by CONACYT-PRONACES/194171/2020 México to VHH. By a grant from VIEP-BUAP to Cuerpo Académico en Neuroendocrinología (CA-BUAP-288) 2023 to JRE and MC"  

4. Please expand the acronym “CONACYT, VIEP-BUAP” (as indicated in your financial disclosure) so that it states the name of your funders in full. This information should be included in your cover letter; we will change the online submission form on your behalf.

"This work was supported by CONACYT-PRONACES/194171/2020 México to VHH. By a grant from VIEP-BUAP to Cuerpo Académico en Neuroendocrinología (CA632 BUAP-288) 2023; and by CONICYT-FONDECYT; Chile to M. Roncagliolo. Thanks to Dr. Yonatan Puón and Dra. Norma Chantal-Seoane for editing the English-language text"

Funding information should not appear in the Acknowledgments section or other areas of your manuscript. We will only publish funding information present in the Funding Statement section of the online submission form. 

"This work was supported by CONACYT-PRONACES/194171/2020 México to VHH. By a grant from VIEP-BUAP to Cuerpo Académico en Neuroendocrinología (CA-BUAP-288) 2023 to JRE and MC"

7. PLOS requires an ORCID iD for the corresponding author in Editorial Manager on papers submitted after December 6th, 2016. Please ensure that you have an ORCID iD and that it is validated in Editorial Manager. To do this, go to ‘Update my Information’ (in the upper left-hand corner of the main menu), and click on the Fetch/Validate link next to the ORCID field. This will take you to the ORCID site and allow you to create a new iD or authenticate a pre-existing iD in Editorial Manager. Please see the following video for instructions on linking an ORCID iD to your Editorial Manager account: https://www.youtube.com/watch?v=_xcclfuvtxQ

Reviewers' comments:

Reviewer's Responses to Questions

**Comments to the Author**

1. Is the manuscript technically sound, and do the data support the conclusions?

Reviewer #1: Partly

Reviewer #2: Yes

Reviewer #3: Yes

2. Has the statistical analysis been performed appropriately and rigorously? 

Reviewer #1: I Don't Know

Reviewer #2: N/A

Reviewer #3: N/A

3. Have the authors made all data underlying the findings in their manuscript fully available?

Reviewer #1: Yes

Reviewer #2: Yes

Reviewer #3: Yes

4. Is the manuscript presented in an intelligible fashion and written in standard English?

Reviewer #1: No

Reviewer #2: Yes

Reviewer #3: Yes

5. Review Comments to the Author

Reviewer #1: The manuscript entitled ‘4-aminopyridine improves evoked potentials and ambulation in the taiep rat: a model of hypomyelination with atrophy of basal ganglia and cerebellum’ by Eguibar et al. aimed to analyze the characteristics of somatosensory evoked potentials (SSEPs) and motor evoked potentials (MEPs) in adult taiep rats. In addition, authors evaluated the effects of 4-aminopyridine on sensory responses and locomotion, also myelin loss in the spinal cord of taiep rat. Authors performed electro recordings at different regions of CNS and compared the velocity of propagation before and after 4-AP application in wildtype and taiep rats. However, as the first sentence of the abstract indicates, the taiep rats have hypomyelination. And hypomyelination reduces the conduction velocity of action potentials as expected. Therefore, it is not clear what is the novel point of this study? In addition, the manuscript, especially the result parts need to be better written. Authors do not explain what their observation indicates. They describe the results but do not interpret these data in deep. Authors may give a short summary or conclusion of each results, and describe what do these changes indicate or suggest. I strongly recommend the authors go through the text carefully and thoroughly to write it in a more professional manner.

In addition, in figure 1, both wt and mutant mice showed similar amplitude, but in figure 3, the amplitude of controls are almonst four times higher than that of the mutant mice without 4-AP treatment. Could the authors explain the discrepancy between the control data in these two figures?

Authors may add tags to indicate N2 in figure 1 or give a scheme of N1 and N2 in figure 2.

In line 452 and 456, fluorescence intensity of what? Myelin or NF?

Reviewer #2: Dear Author(s),

I must commend your work on this well-written article which is a great addition to the body of knowledge. The article gives deep insight that will most likely come in handy in the diagnosis and management of patients with H-ABC. I particularly like that your methods section was detailed and very clear in a manner that can be replicable. Well done once again.

Reviewer #3: lines 80 and 81 : despite it may be true to use fore and hind limb, considering a human standing in a standard anatomical posture, it's good to use lower and upper extremities instead of forelimb and hindlimb.

Please recheck all keywords considering the mesh

PLEASE make a revision about grammatical structures and dictation, for example at the last paragraph of introduction, lines 154 and 155 "the aim of this study we analyze..." there is no agreement between noun and the verb. And in line 450 there is dictation error.

lines 181-184-352: the reference using which you have adjusted the dose of anesthetic drug must be mentioned.

line 185:the size of catheter( thickness and outer diameter) must be mentioned, for instance gauge 22.

lines 187 and 215 :information about product such as manufacturer company should be said.

lines 191 and 192:it's crucial to name or cite the source of information using which you have distinguished the exact region. for example Paxinos atlas or other reefrences.

lines 234 and 235:the model number of the camera should be addressed.

lines 239-246 :The entire paragraph related to the behavioral testing process lacks reference and should be revised. additionally, the measure of room light that can affect rat behavioral factors must have been measured and reported in lux.

lines 250-255: as before, in the paragraph pertaining to Drugs, there was no citation about the references that based on them the drug adjustment was made.

line 268: this section of the article ( Tissue preparation and image acquisition) should go under revision because of the lack of citations about protocol.

line 271: The dosage of the anesthetic used in the anesthesia process before the preparation of the tissue was wrongly mentioned, and usually the dosage of ketamine should be about ten times that of xylazine

At the section" Morphological characteristics of the spinal cord" What protocol and software is used to quantify data related to immunostaining images? Software similar to ImageJ and the like?

There are many outdated references In the Reference section, and are from the 1980s. Please replace with more up-to-date sources.

6. PLOS authors have the option to publish the peer review history of their article (what does this mean?). If published, this will include your full peer review and any attached files.

Reviewer #1: No

Reviewer #2: No

Reviewer #3: **Yes: **Jalil Alizadeh Ghalenoei

---

## [Author Response · Author response to Decision Letter 0]

26 Dec 2023

Reviewer 1.

1. However, as the first sentence of the abstract indicates, the taiep rats have hypomyelination. And hypomyelination reduces the conduction velocity of action potentials as expected. Therefore, it is not clear what is the novel point of this study?.

REPLY: We previously demonstrated a progressive deterioration of conduction velocity in the optic nerve under in vitro conditions (Roncagliolo et al., 2006), and this reduction correlated with the reduction of myelin thickness (Lunn et al., 1997). The present study is the first study in vivo and we add the analysis of the effects of 4-aminopyridine (4-AP) in taiep rats a model of hypomyelination with atrophy of the basal ganglia and cerebellum (H-ABC). Finally, we compare our results of the somatosensory evoked potentials with the first reported Mexican patient with H-ABC.

Our results could be the first step to test 4-AP in human patients with H-ABC or even with others leukodystrophies or leukoencephalopathies as a therapeutic agent.

2. In addition, the manuscript, especially the result parts need to be better written. Authors do not explain what their observation indicates. They describe the results but do not interpret these data in deep. Authors may give a short summary or conclusion of each results, and describe what do these changes indicate or suggest. I strongly recommend the authors go through the text carefully and thoroughly to write it in a more professional manner.

REPLY: We really appreciate the observation did by the reviewer. We rewritten several paragraphs, and all the Ms was reviewed by a native English speaker. We did all experiments rigorously and the corresponding analysis following the best practices in scientific research. But, in base of your suggestion, we review all the Ms carefully in order to improve its quality.

3. REPLY: Thank you for the observation. Now in each section of the Results section we added a short summary to emphasize the experimental data obtained.

4. In addition, in figure 1, both wt and mutant mice showed similar amplitude, but in figure 3, the amplitude of controls are almost four times higher than that of the mutant mice without 4-AP treatment. Could the authors explain the discrepancy between the control data in these two figures?

REPLY: We really appreciate your observation. The results obtained are in rats not in mice. We change Figure 1 to see the distribution of the somatosensory evoked potentials along lumbar and the lowest thoracic spinal cord levels. All evoked potentials had similar amplitudes, only we changed the scale to emphasize our results.

5. Authors may add tags to indicate N2 in figure 1 or give a scheme of N1 and N2 in figure 2. 

REPLY: The observation is quite useful and now we added the suggested tags in the N1 and N2 waves of the somatosensory evoked potentials (SSEPs) on Figure 1 and in Figure 2 as suggested.

We also change the description in the results section and the corresponding Figure legends of both figures.

6. In line 452 and 456, fluorescence intensity of what? Myelin or NF?

REPLY: It is a very useful observation, we appreciate it. Now fluorescence intensity refers to myelin. We rewritten the paragraph to be clearer and it is also the case in Fig. 8 and the corresponding figure legend.

Reviewer 2.

I must commend your work on this well-written article which is a great addition to the body of knowledge. The article gives deep insight that will most likely come in handy in the diagnosis and management of patients with H-ABC. I particularly like that your methods section was detailed and very clear in a manner that can be replicable. Well done once again.

REPLY: We really appreciate his/her comments. We feel motivated to be extremely careful with the methods in future manuscripts.

Reviewer 3.

1. lines 80 and 81: despite it may be true to use fore and hind limb, considering a human standing in a standard anatomical posture, it's good to use lower and upper extremities instead of forelimb and hindlimb.

REPLY: We follow up the suggestion and now we refer to lower and upper extremities along the manuscript.

2. Please recheck all keywords considering the mesh.

REPLY: Thank you for the observation. We change the keyworks to be more assertive and we will have more impact in the scientific community.

3. PLEASE make a revision about grammatical structures and dictation, for example at the last paragraph of introduction, lines 154 and 155 "the aim of this study we analyze..." there is no agreement between noun and the verb. And in line 450 there is dictation error.

REPLY: This is very good observation and I really appreciated it. Now we rewritten the aim of the study and in line 450 we also rewrite the paragraph. We review carefully all the Ms and a native English speaker editing the English language text.

4. lines 181-184-352: the reference using which you have adjusted the dose of anesthetic drug must be mentioned.

REPLY: We added references to adequate level of anesthesia in Material and Method section through all experimental session. I added the paper of Nery, 1968.

5. line 185: the size of catheter (thickness and outer diameter) must be mentioned, for instance gauge 22.

REPLY: Thank you for your observation we added the size of the catheter use for injection of fluids 22 gauge (0.71 mm).

6. lines 187 and 215: information about product such as manufacturer company should be said.

REPLY: The silver electrode was purchased from (WPI, Inc., USA) and we added in both paragraphs as suggested.

7. lines 191 and 192: it's crucial to name or cite the source of information using which you have distinguished the exact region. for example, Paxinos atlas or other references.

REPLY: Thank you for the observation we added a phrase indicating that we follow up the stereotaxic coordinates of the stereotaxic atlas of Paxinos and Watson (2013).

8. lines 234 and 235: the model number of the camera should be addressed.

REPLY: We used a Nikon stereomicroscope model_______ and we draw the area of cortical stimulation in an image of the stereotaxic lamina using in the antero-posterior and medio-lateral coordinates.

9. lines 239-246: The entire paragraph related to the behavioral testing process lacks reference and should be revised. additionally, the measure of room light that can affect rat behavioral factors must have been measured and reported in lux.

REPLY: Thank you for your observation. We added a reference related to novelty-induced ambulation test by Padilla et al., 2010. The intensity of light during this procedure was 149 lux and it was added to the corresponding paragraph.

10. lines 250-255: as before, in the paragraph pertaining to Drugs, there was no citation about the references that based on them the drug adjustment was made.

REPLY: We already had the reference 41 from Peña and Tapia 1999 who made in vivo rats using microdyalisis in the hippocampus during different 4-AP doses. In base of that we used the 1 mg/Kg dose in order to obtain adequate results avoiding but avoiding epilepsy crisis.

11. line 268: this section of the article (Tissue preparation and image acquisition) should go under revision because of the lack of citations about protocol.

REPLY: We added references of the protocol in base of Alata et al., 2021.

12. line 271: The dosage of the anesthetic used in the anesthesia process before the preparation of the tissue was wrongly mentioned, and usually the dosage of ketamine should be about ten times that of xylazine.

REPLY: Thank you for your observation. It was a mistake now the doses of ketamine and xylazine were adjusted.

13. At the section" Morphological characteristics of the spinal cord" What protocol and software is used to quantify data related to immunostaining images? Software similar to ImageJ and the like?

REPLY: Thank you for your observation. Now we added “The protocol used for tissue preparation was described previously (Alata et al., 2021). Briefly three taiep and three WT rats aged three months were anesthetized with a mixture of ketamine/xylazine…”. We used FIJI software to analyze and measure myelin immunostaining. In base on ImageJ but this software offers additional capabilities. We also added the reference Schindelin, 2012.

14. There are many outdated references In the Reference section, and are from the 1980s. Please replace with more up-to-date sources.

REPLY: Thank you for your observation. We did a detail review of the literature. However, there are only a few studies using clinical neurophysiology such as evoked potentials. Probably because now the clinicians preferred MRI and other imageology techniques.

---

## [Decision Letter · Decision Letter 1]

22 Jan 2024

4-aminopyridine improves evoked potentials and ambulation in the taiep rat: a model of hypomyelination with atrophy of basal ganglia and cerebellum.

PONE-D-23-18940R1

Dear Dr. Jose,

We’re pleased to inform you that your manuscript has been judged scientifically suitable for publication and will be formally accepted for publication once it meets all outstanding technical requirements.

Kind regards,

Nejat Mahdieh

Academic Editor

PLOS ONE

Additional Editor Comments (optional):

Reviewers' comments:

Reviewer's Responses to Questions

**Comments to the Author**

1. If the authors have adequately addressed your comments raised in a previous round of review and you feel that this manuscript is now acceptable for publication, you may indicate that here to bypass the “Comments to the Author” section, enter your conflict of interest statement in the “Confidential to Editor” section, and submit your "Accept" recommendation.

Reviewer #3: All comments have been addressed

2. Is the manuscript technically sound, and do the data support the conclusions?

Reviewer #3: Yes

3. Has the statistical analysis been performed appropriately and rigorously? 

Reviewer #3: Yes

4. Have the authors made all data underlying the findings in their manuscript fully available?

Reviewer #3: Yes

5. Is the manuscript presented in an intelligible fashion and written in standard English?

Reviewer #3: Yes

6. Review Comments to the Author

Reviewer #3: I am sincerely grateful to the esteemed authors,especially the corresponding author, for considering my comments and I wish them success in continuing their scientific journey.

7. PLOS authors have the option to publish the peer review history of their article (what does this mean?). If published, this will include your full peer review and any attached files.

Reviewer #3: **Yes: **Jalil Alizadeh Ghalenoei

---

## [Editor Report · Acceptance letter]

21 Feb 2024

PONE-D-23-18940R1 

PLOS ONE

Dear Dr. Eguibar, 

I'm pleased to inform you that your manuscript has been deemed suitable for publication in PLOS ONE. Congratulations! Your manuscript is now being handed over to our production team.

Kind regards, 

on behalf of

Dr. Nejat Mahdieh 

Academic Editor

PLOS ONE